# *cfr* and *fexA* genes in methicillin-resistant *Staphylococcus aureus* from humans and livestock in the Netherlands

Leo M. Schouls [1✉], Kees Veldman [2], Michael S. M. Brouwer [2], Cindy Dierikx[3], Sandra Witteveen[1], Marga van Santen-Verheuvel[1], Antoni P. A. Hendrickx [1], Fabian Landman [1], Paul Hengeveld[3], Bart Wullings[4], Michel Rapallini[4], Ben Wit[5], Engeline van Duijkeren [3✉] & the Dutch MRSA surveillance study group*

## Abstract

**Background** Although the Netherlands is a country with a low endemic level of methicillin-resistant *Staphylococcus aureus* (MRSA), a national MRSA surveillance has been in place since 1989. In 2003 livestock emerged as a major reservoir of MRSA and currently livestock-associated MRSA (clonal complex CC398) make up 25% of all surveillance isolates. To assess possible transfer of resistant strains or resistance genes, MRSA obtained from humans and animals were characterized in detail.

**Methods** The sequenced genomes of 6327 MRSA surveillance isolates from humans and from 332 CC398 isolates from livestock-related samples were analyzed and resistance genes were identified. Several isolates were subjected to long-read sequencing to reconstruct chromosomes and plasmids.

**Results** Here we show the presence of the multi-resistance gene *cfr* in seven CC398 isolates obtained from humans and in one CC398 isolate from a pig-farm dust sample. Cfr induces resistance against five antibiotic classes, which is true for all but two isolates. The isolates are genetically unrelated, and in seven of the isolates *cfr* are located on distinct plasmids. The *fexA* gene is found in 3.9% surveillance isolates and in 7.5% of the samples from livestock. There is considerable sequence variation of *fexA* and geographic origin of the *fexA* alleles.

**Conclusions** The rare *cfr* and *fexA* resistance genes are found in MRSA from humans and animals in the Netherlands, but there is no evidence for spread of resistant strains or resistance plasmids. The proportion of *cfr*-positive MRSA is low, but its presence is worrying and should be closely monitored.

## Plain language summary

A group of bacteria that cause difficult-to-treat infections in humans is methicillin-resistant *Staphylococcus aureus* (MRSA). Monitoring the spread of MRSA strains and genes that cause antibiotic resistance is important for appropriate intervention. In the Netherlands, 25% of MRSA isolates from patients are MRSA types often found in livestock (LA-MRSA). In this study we have identified the *cfr* gene in a small number of LA-MRSA obtained from humans and animals. The *cfr* gene causes resistance to five antibiotic classes, including the last resort antibiotic linezolid. We also found that MRSA from humans and animals carried the antibiotic resistance gene *fexA* and these were often also LA-MRSA. The results suggest that these resistance genes originate from livestock and were transferred to humans. Large scale antibiotic treatment of livestock may lead to increased antibiotic resistance in MRSA found in humans.

[1] National Institute for Public Health and the Environment (RIVM), Infectious Diseases Research, Diagnostics and laboratory Surveillance (IDS), Bilthoven, The Netherlands. [2] Wageningen Bioveterinary Research (WBVR), Bacteriology, Host Pathogen Interaction & Diagnostics, Lelystad, The Netherlands. [3] National Institute for Public Health and the Environment (RIVM), Zoonoses and Environmental Microbiology (Z&O), Bilthoven, The Netherlands. [4] Wageningen Food Safety Research, Team Bacteriology, Molecular Biology & AMR, Wageningen, The Netherlands. [5] Netherlands Food and Consumer Product Safety Authority (NVWA), Food safety, Apeldoorn, The Netherlands. *A list of authors and their affiliations appears at the end of the paper. ✉email: Leo.Schouls@rivm.nl; Engeline.van.Duijkeren@rivm.nl

Due to restricted use of antibiotics and implementation of a so-called Search and Destroy policy the Netherlands is a country with a low endemic level of methicillin-resistant *Staphylococcus aureus* (MRSA). Nevertheless, MRSA cause nosocomial transmissions and community outbreaks and remain a serious healthcare problem. For this reason, national surveillance of MRSA in humans has been implemented in 1989. The surveillance is used to assess changes in characteristics of MRSA, including antibiotic resistance. During the last decades livestock has emerged as a major source for MRSA colonizing and infecting humans in the Netherlands. These CC398 MRSA, designated as livestock-associated MRSA (LA-MRSA), currently make up 25% of all isolates submitted in the surveillance. In Europe and North America, LA-MRSA is dominated by the clonal complex CC398, whereas in Asia CC9 is the dominant LA-MRSA clonal complex[1]. Antibiotic resistance of MRSA obtained from humans in the Netherlands is dominated by resistance to antibiotic classes like tetracyclines, aminoglycoside, macrolides, lincosamides, ciprofloxacin and trimethoprim, with proportions ranging from 22% to 43%. During the last decades new resistance genes such as the chloramphenicol-florfenicol resistance gene (*cfr*), which encodes resistance to phenicols, lincosamides, oxazolidinones, pleuromutilins, and streptogramin A (PhLOPS$_A$), and the florfenicol exporter gene (*fexA*) have been discovered[2,3]. Finding such rarely occurring resistance traits may indicate import and spread of more resistant MRSA and therefore more difficult to treat MRSA infections. For this reason, we have studied the occurrence and nature of these resistance genes in MRSA obtained from humans and animals in the Netherlands.

The study shows the *cfr* multi-resistance gene was found in CC398 strains from humans and animals at very low frequency and that we found no evidence for spread of a resistant strain or a *cfr* resistance plasmid. The heterogenic resistance gene *fexA* was found more frequently, but almost exclusively in CC398 from humans and livestock and in CC5 MRSA from humans. The results suggest that MRSA in livestock may act as a reservoir for transfer of antibiotic resistance genes to MRSA found in humans.

## Methods

**Bacterial isolates and specimens**. For the Dutch national MRSA surveillance, medical microbiology laboratories (MMLs) in the Netherlands send isolates from MRSA carriers and from persons infected with MRSA to the National Institute for Public Health and the Environment (RIVM). Since 2008 the RIVM received and typed 53,048 MRSA isolates obtained from humans[4]. A subset of these isolates was sequenced for various research projects and thus this collection is incomplete not a random set. However, the collection also contained all MRSA isolates (*n* = 1986, one isolate per person) received in the second quarter (Q2) of 2019, Q2 of 2020 and Q4 of 2020, making it a complete 9-month surveillance collection. In total, NGS data of 6327 MRSA isolates obtained from humans were used in the study (Supplementary Table 1).

The veterinary MRSA collection used in this study comprised 332 sequenced CC398 isolates originating from various samples of livestock, dust samples from farms, nasal swabs from persons working on these farms, and retail meat collected in different studies between 2001–2019 (Supplementary Table 2). In this study these isolates are referred to as the MRSA isolates of the livestock sampling.

**Metadata**. MMLs provided the sampling date, the nature of the specimen, the type of health-care provider, gender, age in years, four digits of the postcode and a pseudonymized person identifier. Since the introduction of the digital data exchange Type-Ned system for MRSA surveillance in November 2016, MMLs and

infection prevention workers filled out digital questionnaires to provide additional data on persons and to assess risks factors associated with MRSA infection and colonization. The questionnaires contained questions on the health-care provider, patient's residence, and risk factors for MRSA carriage such as underlying disease, visiting other countries, being hospitalized abroad, and animal contact. Of the 3246 isolates that were sequenced during the 2019–2021 interval, completed questionnaires were obtained for 2752 (85%) of the isolates. Thirteen percent of the patients reported livestock contact, 61% reported no livestock contact and for 26% of the patients livestock contact was unknown. For persons carrying CC398 MRSA 51% (316/624) reported livestock contact and of all persons reporting contact with livestock 90% (316/351) carried CC398 MRSA.

**Ethics statement**. The bacterial isolates belong to the MMLs participating in the Dutch National MRSA Surveillance and were obtained as part of routine clinical care in the past years. Only data on isolate and patient available in the digital Type-Ned system were used in this study. To ensure privacy, person identifiers were pseudonymized before storage in the Type-Ned database. Furthermore, only patient's age in years (not birthdate) and a residential region identifier based the four digits of the zip code only was stored. Only MRSA isolates and not clinical specimens obtained from patients were available and used for this study. Since no identifiable personal data were collected and data were analyzed and processed anonymously, written, or verbal patient consent was not required. According to the Dutch Medical Research Involving Human Subjects Act (WMO) this study was therefore exempt from review by an Institutional Review Board.

**Next-generation sequencing and third-generation sequencing**. MRSA isolates were subjected to next-generation sequencing (NGS) using the Illumina MiSeq and HiSeq 2500[5]. For third-generation (long-read) sequencing, high molecular weight DNA was isolated using an in-house developed protocol[5]. The Oxford Nanopore protocol SQK-RBK004 (https://community.nanoporetech.com) was used in runs of 12 barcoded isolates on a MinION flow cell (MIN-106 R9.4.1). A 48-h sequence run was started on a GridION with live base calling (high accuracy protocol) enabled inside the MinKNOW GUI. De-multiplexing was performed afterwards using Guppy barcoding software version 3.5.1. Read lengths <5000 base pairs were omitted using NanoFilt 2.2.0, subsequently both sides were trimmed 80 bases using head crop and tail crop settings. Additionally, FiltLong 0.2.0 was used to filter for the reads with the 90% highest score and make a subset up to a maximum of 500 Mb. To reconstruct plasmids and chromosomes, Illumina and nanopore data were used in a hybrid assembly performed in Unicycler v0.4.4[6]. The plasmids and chromosomes obtained were annotated using Prokka[7]. All assembled plasmids and chromosomes are deposited in the NCBI database under the following accession numbers: H1_RIVM_M044329: CP096540-CP096541, H2_RIVM_M047065: CP096539, H3_RIVM_M047916: CP096535-CP096538, H4_RIVM_M083782: CP096532-CP096534, H5_RIVM_M084526: CP096530-CP096531, H6_RIVM_M084986: CP096528-CP096529, H7_RIVM_M087195: CP096526-CP096527, P1_RIVM_M085090: CP096522-CP096525. In the number H1_RIVM_M044329 the H1 is a short notation for the first isolate from humans in this study, and RIVM_M044329 represents the unique identifier used for this isolate in the RIVM database.

**Molecular analyses**. NGS data were imported into CLC Genomics Workbench (Version 21.0.4, QIAGEN Aarhus A/S) and used in de novo assemblies to generate contigs. To assign

genogroups and assess the genetic relationship between isolates, the contigs were used for wgMLST analyses using the COL-based wgMLST scheme available via the SeqSphere software version 6.0.2 (Ridom GmbH, Münster, Germany)[8]. Allelic profiles were imported into BioNumerics version 7.6.3 (Applied Maths, Sint-Martens-Latem, Belgium) for subsequent comparative analyses. Other manipulations of the NGS data such as read mapping, alignments, local BLAST analyses etc. were performed in CLC Genomics Workbench.

The antibiotic resistance gene profiles of acquired resistance genes and chromosomal mutations mediating antimicrobial resistance were assessed using software and databases downloaded from the Center for Genomic Epidemiology[9–11]. Insertion elements were identified using IS-Finder (https://isfinder.biotoul.fr/) and for identification of tandem repeats the Tandem Repeats Finder program was used[12].

**Antibiotic Susceptibility Testing**. Antibiotic susceptibility testing (AST) was performed with broth microdilution according to ISO standards (ISO 20776-1:2019) using a European panel (EUST) designed for testing staphylococci consisting of 19 different antibiotics (Sensitire©, Trek Diagnostic Systems, UK). In addition, a second, custom made panel (NLD1GNS) with seven additional antibiotics, was included. For interpretation of the results European panel epidemiological cut-off values (ECOFFs) were used as recommended by the European Committee on Antimicrobial Susceptibility Testing (EUCAST; http://mic.eucast.org). When ECOFFs were lacking, animal-specific clinical breakpoints from the Clinical and Laboratory Standards Institute (CLSI, VET08) were used to interpret the MICs.

**Reporting summary**. Further information on research design is available in the Nature Research Reporting Summary linked to this article.

## Results

**Isolates carrying the *cfr* gene**. The 6327 isolates obtained from humans comprised 1641 CC398 isolates and 4686 non-CC398 isolates. Seven isolates from seven patients (H1-H7) carried the rare *cfr* multi-resistance gene and all *cfr*-positive isolates were CC398 (ST398). In addition, one LA-MRSA isolate (P1) of the livestock sampling set, obtained from a dust sample collected in a pig farm, also carried the *cfr* gene. All persons with *cfr*-positive MRSA lived in the mid-eastern part of the Netherlands (Fig. 1). Four of the seven persons carrying these *cfr*-positive isolates reported having professional contact with livestock (pigs), one person claimed not to have been in contact with livestock and for two persons livestock contact was unknown. Based on the data of all sequenced national surveillance isolates obtained in a 9-month period (2019-Q2, 2020-Q2 and 2020-Q4) we estimate that 0.2% (4/1986) of the isolates submitted for the Dutch MRSA.

The Cfr protein methylates the 23 S rRNA molecules rendering the bacterium resistant to five different antibiotic classes: phenicols, lincosamides, oxazolidinones, pleuromutilins and streptogramin A, the so-called PhLOPS$_A$ phenotype. To assess whether the *cfr* gene in the eight CC398 isolates indeed caused this multi-resistance phenotype, AST was performed (Table 1, Supplementary Table 3). The multi-resistance phenotype was found in only six of the eight *cfr*-positive isolates. Closer inspection revealed that the *cfr* gene in H5 carried a mutation and in H7 the *cfr* gene had a single base pair deletion (Supplementary Fig. 1). Both changes caused a premature termination of the translation of the gene corroborating the phenotypic susceptibility testing results.

**Genetic relationship of isolates**. To assess whether the *cfr*-positive isolates were genetically related and represented an outbreak or spread of a *cfr*-positive strain wgMLST was performed. This showed that they were genetically unrelated (Fig. 2). Pairwise comparison of the *cfr*-positive isolates showed that H2 and H3 were the closest related isolates with an allelic distance of 41 alleles (Supplementary Fig. 2). The most distantly related isolates were H6 and P1.

**Genetic organization of *cfr* and *fexA* genes**. All *cfr*-positive isolates were also subjected to long-read sequencing and hybrid assembly was used to reconstruct their chromosomes and plasmids. This revealed that in all, except one isolate, the *cfr* gene was located on a plasmid (Table 2, Supplementary Table 4). The *cfr* plasmid sizes ranged from 14 kbp to 40 kbp. All isolates also carried a *fexA* gene which in four isolates was located on the same plasmid as the *cfr* gene and in four isolates on the chromosome.

The reconstructed plasmids and chromosomes showed a diverse and complex organization of the *cfr* and *fexA* genes (Fig. 3). In H1 the *cfr* gene was located on a small plasmid in which the *cfr* gene region was flanked by two copies of IS*431*mec, and the *fexA* gene was located on the chromosome. Of note is that the MIC for linezolid in H1 was elevated but did not reach the level of resistance. In H2 the *cfr* gene appeared to be integrated in the region on the chromosome approximately at position 1.7–1.8 Mbp, where the *radC* gene normally resides. In the other isolates the *cfr* gene was located together with *fexA* on the same plasmid. The plasmid of H4 may be the product of recombination of various sources due to various insertion elements such as the three copies of the IS*431*mec insertion element, the IS*21* genes *istA* and *istB* and the Tn*558* *tnpA*, *tnpB* and *tnpC* genes. All eight isolates contained complete or truncated genes of the Tn*558* transposon, always associated with the *fexA* gene. H3 carried a *fexA14* variant gene on both the plasmid and the chromosome. Similarly, the *dfrK* gene was present on a plasmid and in the chromosome of H4.

There was a close resemblance between the *cfr-fexA* plasmids of H3, H5 and P1 (Fig. 3). The plasmids of H5 and P1 were nearly identical with a 95% sequence identity. However, the 9263 bp segment Tn*558*.3R-*fexA-cfr*-Tn*558*L of P1 was inverted compared to H5. These segments were nearly identical except for three point mutations. The *cfr-fexA* plasmids of H3, H5, and P1 also closely resembled the *S. aureus* *cfr-fexA* plasmid deposited under accession number CP065195 (>88% identity). Latter plasmid was from a CC398 isolate obtained on a pig farm in China in 2016. The *cfr-fexA* plasmids of H3 and CP065195 were the most closely related plasmids (>98% identity), both carrying the IS*21* genes *istA* and *istB* in the *cfr-fexA* region. Remarkably, the *cfr* gene in CP065195 had the same single base pair deletion as the *cfr* gene of H7 and has been annotated as a pseudo gene. Interrogation of the NCBI database revealed that this defective *cfr* gene was identified by others in two more CC398 isolates, one obtained from a pig in Italy (chromosome, MW298531) and the other from a pig in Australia (plasmid, CP029172).

The *cfr-fexA* plasmids of H3, H5 and P1 carried a tandem repeat region with 35 bp long perfect tandem repeats with the sequence TATGTGAGAAGATATATAGAGTATATGCAACTGTA. The H3 plasmid carried 43 repeats, the H5 plasmid 37 repeats and the plasmid in P1 carried 17 repeats. The repeats were flanked by sequences that were identical in the three isolates. The CP065195 plasmid also carried 39 tandem repeats in this region of the plasmid, albeit with a different sequence (CTCAATATATCTTCTTACATATGCTTATATATATA) and with a quarter of the repeats having an imperfect repeat sequence. In all four plasmids the repeats are flanked by two open reading frames. The

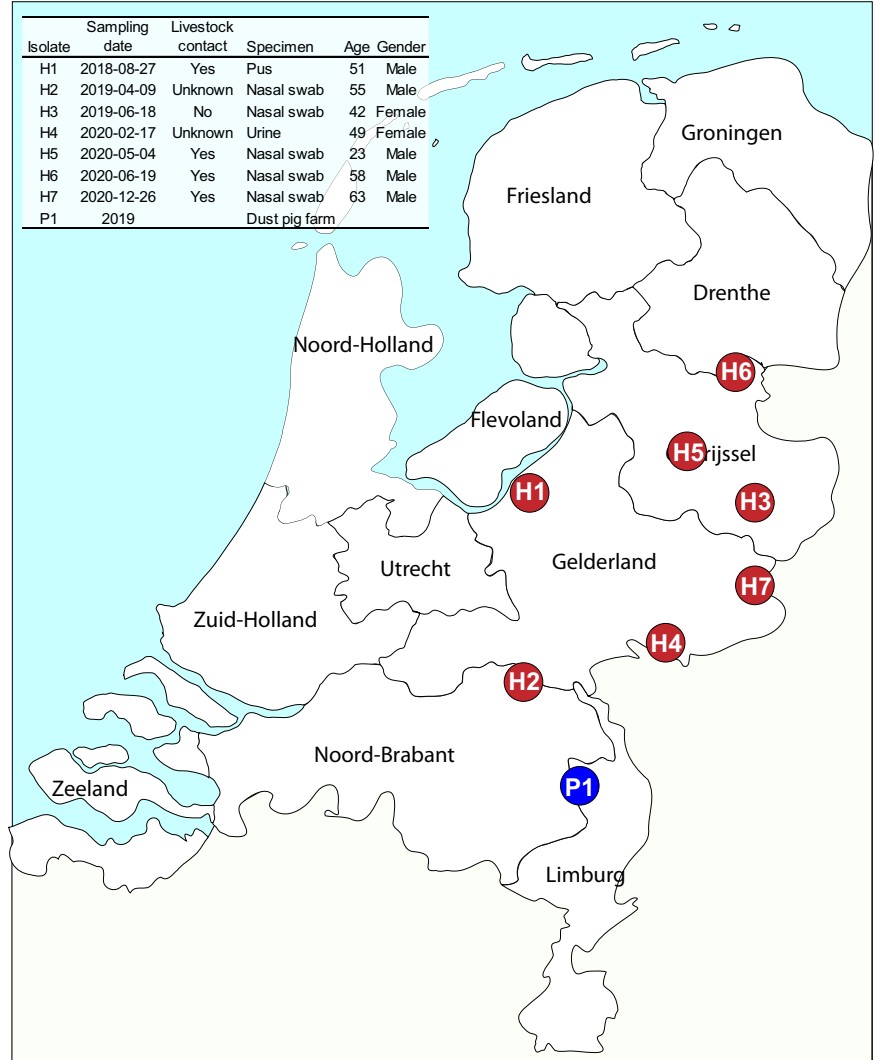

| Isolate | Sampling date | Livestock contact | Specimen | Age | Gender |
|---|---|---|---|---|---|
| H1 | 2018-08-27 | Yes | Pus | 51 | Male |
| H2 | 2019-04-09 | Unknown | Nasal swab | 55 | Male |
| H3 | 2019-06-18 | No | Nasal swab | 42 | Female |
| H4 | 2020-02-17 | Unknown | Urine | 49 | Female |
| H5 | 2020-05-04 | Yes | Nasal swab | 23 | Male |
| H6 | 2020-06-19 | Yes | Nasal swab | 58 | Male |
| H7 | 2020-12-26 | Yes | Nasal swab | 63 | Male |
| P1 | 2019 | | Dust pig farm | | |

**Fig. 1 Characteristics of the persons who carried *cfr*-positive MRSA.** The location of the residences of the seven persons carrying *cfr*-positive CC398 isolates are displayed as red circles on the geographic map of the Netherlands. The pig farm from which the *cfr*-positive isolate originated is indicated by the blue circle. The table in the inset shows some characteristics of the patients and specimens.

**Table 1 Minimum inhibitory concentrations against PhLOPS$_A$ antibiotics.**

| | Phenicol | | Lincosamide | Oxazolidinone | Pleuromutilin | Streptogramin A |
|---|---|---|---|---|---|---|
| Isolate | Chloramphenicol (16) | Florfenicol (8) | Clindamycin (0.25) | Linezolid (4) | Tiamulin (2) | Quinupristin/Dalfopristin (1) |
| H1 | 64 | >32 | >4 | 4 | >4 | >4 |
| H2 | >64 | >32 | >4 | 8 | >4 | 4 |
| H3 | >64 | >32 | >4 | 8 | >4 | >4 |
| H4 | >64 | >32 | >4 | 8 | >4 | 2 |
| H5 | 64 | >32 | ≤0.12 | ≤1 | 1 | ≤0.5 |
| H6 | >64 | >32 | >4 | 8 | >4 | >4 |
| H7 | >64 | >32 | ≤0.12 | 2 | ≤0.5 | ≤0.5 |
| P1 | >64 | >32 | >4 | 8 | >4 | 2 |

The top row indicates the antibiotic class and the second row the antibiotic. The value in parenthesis indicates the EUCAST epidemiological cut-off values (R: >) in mg/L.

first encodes for a primase C-terminal domain-containing protein, the second encodes for a DUF87 protein. The *cfr-fexA* plasmid of H4 carried 11 perfect tandem repeats that were 63 bp long with sequence AGTAGAATATACTACTTATGTCTTTTTC TATTATTCTACATGACTACTTAACTACTCATTTAT, but was not flanked by the two genes found in H3, H5, P1 and the CP065195 plasmid.

**The heterogenic *fexA* gene.** All eight *cfr*-positive isolates also carried the *fexA* gene. Analysis of the complete study collection revealed that 246 of the 6327 MRSA surveillance isolates (3.9%) and 25 of the 332 MRSA isolates from the livestock sampling (7.5%) carried a *fexA* gene. Based on the collection of MRSA isolates obtained in the 9-month period in which we sequenced all received isolates, we estimate that 5.7% (114/1986) of the

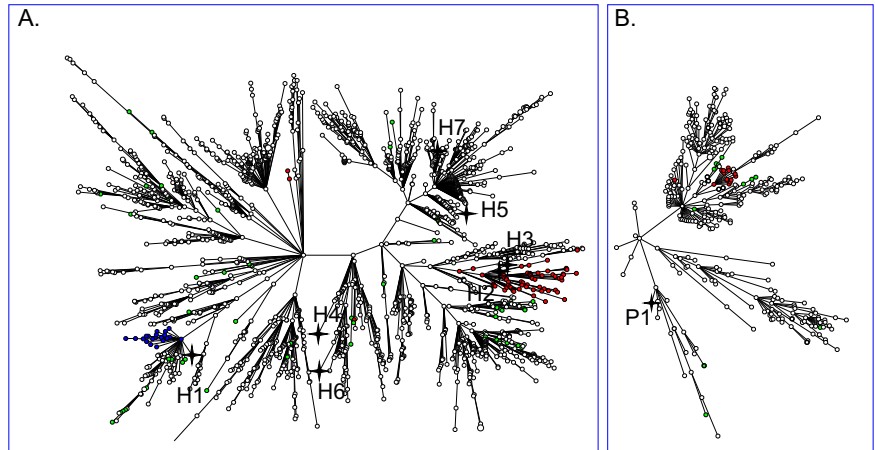

**Fig. 2 wgMLST-based genetic relationship of CC398 isolates carrying the *cfr* gene. A** CC398 isolates obtained from the Dutch national MRSA surveillance, and all were obtained from humans. **B** MRSA isolates obtained in the livestock sampling. The *cfr*-positive isolates are marked by a four-pointed star. The red circles denote *fexA03*-positive isolates, blue circles *fexA05* and green circles other *fexA* variants. Isolates obtained from humans working on animal farms or slaughterhouses are depicted as circles with a thick line.

**Table 2 Resistance genes in chromosomes and plasmids of *cfr*-positive isolates.**

| Genomes | Size (bp) | cfr | fexA variant | aadD | ant(6)-Ia | ant(9)-Ia | dfrG | dfrK | erm(B) | erm(C) | lnu(A) | lnu(B) | lsa(B) | lsa(E) | str | tet(K) | tet(L) | tet(M) |
|---|---|---|---|---|---|---|---|---|---|---|---|---|---|---|---|---|---|---|
| H1 chromosome | 2,933,086 | | fexA19 | ■ | ■ | | ■ | | | | | ■ | | ■ | | ■ | | ■ |
| H1 plasmid 1 | 13,757 | ■ | | | | | | | | ■ | | | | | | | ■ | |
| H2 chromosome | 2,806,671 | ■ | fexA20 | | | | | | | | | | | | | ■ | | ■ |
| H3 chromosome | 2,844,988 | | fexA14 | | | | | | | | | | | | | ■ | | ■ |
| H3 plasmid 1 | 38,430 | ■ | fexA14 | | | | | | | | | | | | | | | |
| H3 plasmid 2 | 2,997 | | | | | | | | | | | | | | | | | |
| H3 plasmid 3 | 2,407 | | | | | | | | | | ■ | | | | | | | |
| H4 chromosome | 2,893,482 | | | | | | | ■ | | | | | | | | ■ | | ■ |
| H4 plasmid 1 | 39,628 | ■ | fexA21 | ■ | | | | ■ | | | | | ■ | | | | ■ | |
| H4 plasmid 2 | 2,361 | | | | | | | | | | | ■ | | | | | | |
| H5 chromosome | 2,890,216 | | | | | | | ■ | | | | | | | | | | ■ |
| H5 plasmid 1 | 37,004 | ▥ | fexA23 | | | | | | | | | | | | | | | |
| H6 chromosome | 2,795,636 | | | | | | | ■ | | | | | | | | ■ | | ■ |
| H6 plasmid 1 | 35,612 | ■ | fexA20 | | | | | | | | | | | | | | | |
| H7 chromosome | 2,817,149 | | | | | | | ■ | | | | | | | | | | ■ |
| H7 plasmid 1 | 34,523 | ▥ | fexA13 | | | | | | | | | | | | | | | |
| P1 chromosome | 2,796,375 | | | | | ■ | ■ | | | | | | ■ | | ■ | | ■ | | ■ |
| P1 plasmid 1 | 37,279 | ■ | fexA11 | | | | | | | | | | | | | | | |
| P1 plasmid 2 | 4,397 | | | | | | | | | | | | | | ■ | | | |
| P1 plasmid 3 | 3,971 | | | | | | | | | | | | | | | | | |

All isolates carried a *mecA* and a *blaZ* gene in their chromosomes. The black boxes indicate the presence of the resistance genes. The gray boxes indicate the gene is inactive due to mutation or deletion of a single nucleotide.

isolates submitted for the Dutch MRSA surveillance carry a *fexA* gene. Approximately 61% (69/114) of these *fexA*-positive isolates belong to clonal complex CC398 and 37% belong to CC5 (Table 3). The remaining six *fexA*-positive isolates belonged to four different CCs. All, except one, *fexA*-positive isolates from the MRSA from the livestock sampling originated from a porcine source. The exception was an isolate obtained from a human working at a poultry farm.

In the majority of the 271 *fexA*-positive isolates, the gene was located in Tn*558*, accompanied by the characteristic Tn*558* family ends, Tn*558*L and Tn*558*.3R [https://tnpedia.fcav.unesp.br/index.php/Transposons_families/Tn554_family]. Exceptions were the *cfr*-positive isolates, and two isolates from the MRSA surveillance with an incomplete transposon sequence. The *fexA*-carrying Tn*558* transposon in 251 of the remaining 263 isolates was inserted into the *radC* gene in the same orientation. In all of these isolates the *fexA* gene was located on large contigs (>500 kbp), indicative of the chromosomal location. In two isolates the *fexA*-containing contig was small and did not contain the complete transposon and one isolate lacked the upstream region of the

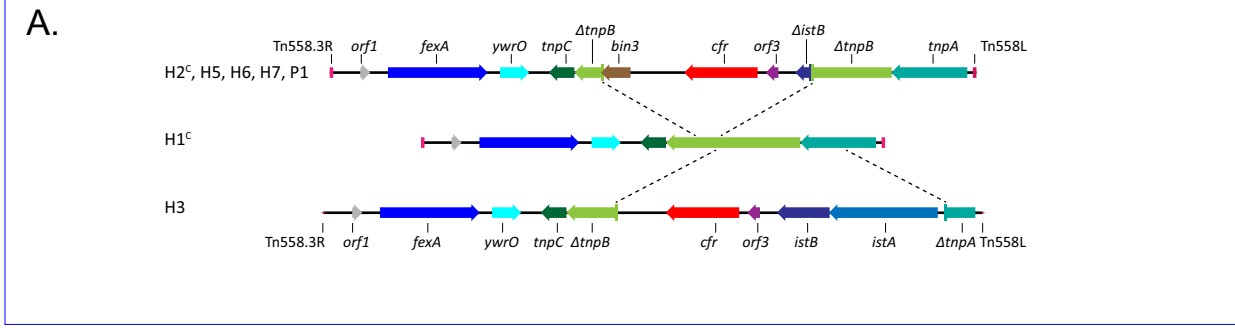

**Fig. 3 Genetic organization of the *cfr* and *fexA* gene environment in eight *cfr*-positive CC398 isolates. A** Insertion of the two predominant *cfr* segment types into Tn*558*. H1^C and H2^C indicate that the segment resides in the chromosome. **B** Composition of the plasmids containing *cfr* (H1) or *cfr* and *fexA* (H2-H7, P1). Relevant genetic elements are colored, orange arrows denote other antibiotic resistance genes, and black arrows denote other putative genes. The blue arrows in the figure with the nested H3, H5 and P1 plasmids, indicate the flipped Tn*558*-region in P1.

**Table 3 Distribution of *fexA* alleles among the MRSA isolates from the national MRSA surveillance and the livestock MRSA sampling.**

| *fexA* variant | National MRSA surveillance | | | | | | | Isolates from livestock sampling | | | | |
| --- | --- | --- | --- | --- | --- | --- | --- | --- | --- | --- | --- | --- |
| | GG0005 (n = 904) | GG0006 (n = 230) | GG0008 (n = 804) | GG0009 (n = 15) | GG0059 (n = 138) | GG0398 (n = 1641) | All GGs (n = 6327) | Human nose (n = 14) | Pig farm dust (n = 58) | Pig meat (n = 14) | Pig nose (n = 14) | All isolates (n = 332) |
| *fexA01* | | | | | | 10 | 10 | | 1 | | 3 | 4 |
| *fexA03* | | | | | | 70 | 70 | | 8 | 2 | 3 | 13 |
| *fexA04* | | | | | | 1 | 1 | | | | | |
| *fexA05* | | | | | | 20 | 20 | | | | | |
| *fexA06* | | | | | | 3 | 3 | | | | | |
| *fexA07* | | | | | | 2 | 2 | 1 | | | | 1 |
| *fexA08* | | | | | | 2 | 2 | | | | | |
| *fexA09* | | | | | | 1 | 1 | | | | | |
| *fexA10* | | | | | | 1 | 1 | | | | | |
| *fexA11* | | | | | | 5 | 5 | | 1 | | | 1 |
| *fexA12* | | 1 | | | | | 1 | | | | | |
| *fexA13* | | | | | | 8 | 8 | | 1 | | | 1 |
| *fexA14* | | | | | | 1 | 1 | | | | | |
| *fexA15* | | | | 1 | | | 1 | | | | | |
| *fexA16* | | | | | | 3 | 3 | | | | 3 | 3 |
| *fexA17* | 79 | | | | | | 79 | | | | | |
| *fexA19* | | | | | | 10 | 10 | | 2 | | | 2 |
| *fexA20* | 1 | | | | | 2 | 2 | | | | | |
| *fexA21* | 2 | | | | | 1 | 1 | | | | | |
| *fexA22* | 1 | | | | | 1 | 1 | | | | | |
| *fexA23* | 3 | | | | | 8 | 8 | | | | | |
| *fexA24* | 1 | | | | | | 1 | | | | | |
| *fexA25* | 2 | | | | | | 2 | | | | | |
| *fexA27* | 1 | | | | | | 1 | | | | | |
| *fexA28* | 3 | | | | | | 3 | | | | | |
| *fexA29* | 1 | | | | | | 1 | | | | | |
| *fexA30* | | | | | 1 | | 1 | | | | | |
| *fexA31* | 1 | | | | | | 1 | | | | | |
| *fexA32* | 1 | | | | | | 1 | | | | | |
| *fexA33* | 2 | | | | | | 2 | | | | | |
| *fexA34* | | | 3 | | | | 3 | | | | | |
| All *fexA* variants | 91 | 1 | 3 | 1 | 1 | 149 | 246 | 1 | 13 | 2 | 9 | 25 |

The isolates from the pig fecal samples (n = 11), cattle samples (n = 22) and poultry samples (n = 78) did not carry a *fexA* gene.

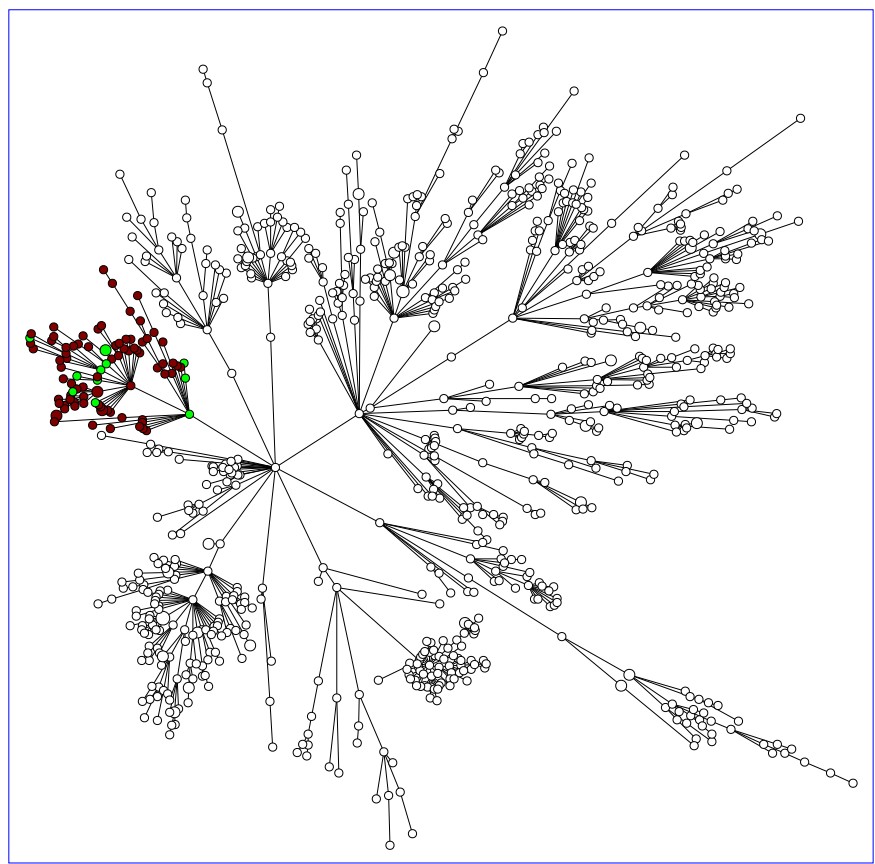

**Fig. 4 wgMLST-based minimum spanning tree of CC5 isolates (*n* = 904).** Brown circles indicate CC5 isolates carrying the *fexA17* allele (*n* = 79), green circles indicate CC5 isolates carrying other *fexA* variants (*n* = 12).

*radC* gene. In eight isolates the Tn*558* was located on contigs varying between 34–42 kbp in size and was not inserted into the *radC* gene. This suggests that in these isolates the *fexA* was located on a plasmid. There was a single isolate from the livestock collection in which the complete Tn*558* was inverted. The Tn*558* was inserted at the hexanucleotide sequence TACTCA (GATGTA inverse complement), which is part of the *radC* sequence and is duplicated due to the insertion. In the eight *cfr*-positive isolates from our study the nature of the hexanucleotide insertion site was variable (Supplementary Fig. 3). In isolates carrying a *dfrK* gene in Tn*559*, the transposon was also inserted in *radC* at the same site.

There was considerable heterogeneity among the *fexA* genes in the isolate collection. Only 14 of the 271 isolates had a *fexA* gene with a sequence that was identical to that of the reference sequence *fexA*_1 used in ResFinder (NCBI acc. num. AJ549214). The ResFinder *fexA*_2 variant (acc. num. AM408573) was not found. Analysis revealed that the collection harbored 30 sequence variants of the *fexA* gene encoding for 27 allelic variants of the FexA protein (Table 3, Supplementary Fig. 4). The *fexA* sequence variants were given provisional names *fexA03* - *fexA34*.

The *fexA03* allele was the predominant allelic variant found exclusively in *fexA*-positive CC398 isolates, both in the national surveillance isolates (47%, 70/149) and the isolates of the livestock sampling (52%, 13/25). Among the national surveillance isolates *fexA05* was the second most dominant allele (13%, 20/149). However, this allele was not found among the isolates of the livestock sampling. All alleles found in the livestock surveillance isolates were also found in the CC398 national surveillance isolates. Of interest, each of the eight *cfr*-positive isolates from this study had a different *fexA* allele (Table 2). In CC5 isolates

*fexA17* was the dominant allele (87%, 79/91) and this allele was not found in CC398 or the other genomic groups. In fact, there was a strict association between clonal complexes and *fexA* allele (Table 3, Supplementary Table 5). Interrogation of the NCBI database revealed that there we 257 entries from various bacterial genera carrying a complete *fexA* gene yielding 42 more *fexA* sequence variants, confirming the high degree of diversity of this gene (Supplementary Tables 6, 7).

The *fexA* alleles were not randomly distributed over the MRSA population. For CC398 the *fexA03* and *fexA05* alleles were predominantly found in certain branches of the wgMLST minimum spanning trees (Fig. 2). For CC5 this was even more extreme as all *fexA*-positive isolates are grouped in a single branch of the tree (Fig. 4).

The geographic distribution of the residences of the persons from whom the *fexA*-positive isolates were obtained appeared not to be random (Fig. 5). It showed that CC398 isolates carrying the *fexA03* allele were predominantly isolated from persons living in the south-eastern part of the country, whereas isolates with *fexA05* were obtained from people living in the mid-eastern part. The residences of persons from whom CC398 MRSA with other *fexA* alleles were isolated, were scattered throughout the mid-eastern and south-eastern part of the country. This is also the region where most of the persons from whom CC398 MRSA are isolated are living and the region with the highest density of livestock farms in the Netherlands. The geographic location of residences of persons from whom CC5 isolates carrying the *fexA* gene were obtained was predominantly to the western part of the country.

**Allelic variation of *fexA* and antibiotic resistance**. The MICs of 24 isolates obtained from humans and animals, comprising *fexA*

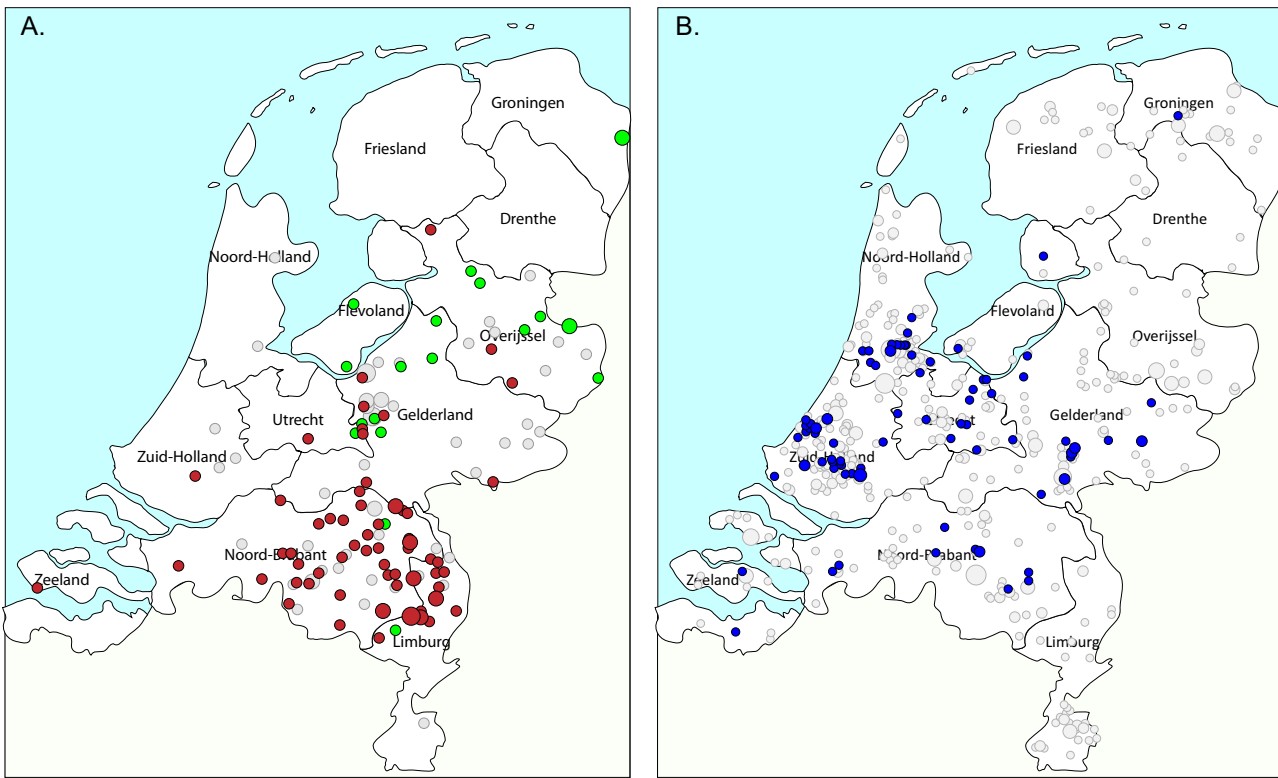

**Fig. 5 Geographic distribution of the residences of persons from whom *fexA*-positive MRSA were isolated. A** Distribution of geolocation persons carrying *fexA*-positive CC398 isolates (*n* = 149); red circles indicate residences of persons carrying *fexA03* CC398 (*n* = 70), green circles *fexA05* (*n* = 20) and gray circles other *fexA* variants (*n* = 59). **B** Distribution of *fexA*-positive (*n* = 91, blue circles) and *fexA*-negative (*n* = 813, gray) CC5 isolates.

variants *fexA01*, *fexA03*, *fexA05*, *fexA17*, *fexA19* and *fexA28* were determined (Supplementary Table 8). This revealed that all isolates were resistant to both chloramphenicol and florfenicol, except for those carrying *fexA17* and *fexA28* which were resistant to chloramphenicol but sensitive for florfenicol. Latter *fexA* alleles were found in CC5 isolates only and not in CC398.

## Discussion

In this study we showed that the rare resistance genes *cfr* and the *fexA* are present in MRSA isolates obtained from humans in the Dutch national MRSA surveillance and in MRSA isolates obtained from livestock samples, livestock environment and from persons with professional livestock contact. In total we found eight isolates carrying the *cfr* gene which all were CC398 (LA-MRSA) isolates. Seven isolates were obtained from humans in the national MRSA surveillance and one isolate originated from a dust sample in a pig farm. The occurrence of the *cfr* gene in MRSA isolates obtained in the Dutch MRSA surveillance is very rare with an estimated proportion of 0.2%. The *cfr*-carrying isolates were genetically unrelated as assessed by wgMLST. In seven of the eight *cfr*-positive isolates the gene was located on plasmids which all differed in genetic composition and in two isolates the *cfr* gene was inactive due to a mutation or deletion. All *cfr*-positive isolates in this study also carried the *fexA* gene, but latter gene was also found in isolates without *cfr*. In MRSA isolates from humans the *fexA* gene was almost exclusively found in clonal complexes CC398 and CC5 at a low estimated proportion of 5.7%. There was a high degree of sequence diversity of the *fexA* gene in the isolates studied and there was a strong association between *fexA* alleles and MRSA clonal complexes. Also, there was a relationship between the location of the residency of *fexA*-positive MRSA carriers and the *fexA* allele found.

The *cfr* gene renders *S. aureus* simultaneously resistant against five different antibiotic classes: phenicols, lincosamides, oxazolidinones, pleuromutilins and streptogramin A (PhLOPS$_A$) through the methylation of the 23 S rRNA[2,13,14]. The *fexA* gene is responsible for the active export of the phenicol antibiotics florfenicol and chloramphenicol[3,15]. In the Netherlands most of these antibiotics are not used for treatment in humans. However, the lincosamide antibiotic clindamycin is being used and outpatients received 0.23 defined daily doses (DDD)/1000 inhabitant-days of lincosamides in 2020, and 2.35 DDD/100 patient-days of lincosamides for systemic use in hospitals in 2020 [NethMap/MARAN 2022, http://hdl.handle.net/10029/625885]. In comparison, the total outpatient use of systemic antibiotics in 2020 was 7.61 DDD/1000 inhabitant days and 85.79 DDD/100 patient-days in hospitals. Chloramphenicol is rarely used, mostly in eye droplets or eye ointments. In the Netherlands and our neighboring countries, the oxazolidinone antibiotic linezolid is classified as a last resort antibiotic, only to be used in specific cases such vancomycin-resistant enterococci[16]. Nevertheless, the World Health Organization (WHO) has declared linezolid as a critically important antimicrobials for human medicine [Critically important antimicrobials for human medicine, 6th revision, WHO. ISBN 978-92-4-151552-8]. The WHO also classified linezolid a Group A drug for treatment of MDR and XDR tuberculosis, to be included in the treatment regimen unless contraindicated. In the Netherlands, only few cases of infection in humans with MDR MRSA or enterococci that require treatment with linezolid or other oxazolidinones occur. Despite the limited use of linezolid in humans the emergence of *cfr*-positive MRSA is worrying. However, the situation is different for infections in livestock where *cfr*-carrying MRSA does not pose a direct risk for animal health, but the potential spread of *cfr* genes to specific animal pathogenic bacteria would limit the options for treating infected animals with antibiotics belonging to veterinary important

antibiotic classes like lincosamides (lincomycin, pirlimycin), pleuromutilins (tiamulin, valnemulin) and phenicols. In the Netherlands, phenicols are mainly used in pigs and veal calves, and only small amounts are used in dairy cattle and no use is reported in poultry (Report of the Netherlands Veterinary Medicines Institute (SDa)): Usage of antibiotics in agricultural livestock in the Netherlands in 2020 (https://www.autoriteitdiergeneesmiddelen.nl/en/). In the present study, most isolates from animals were from pigs, and much less from dairy cattle and poultry, no isolates from veal calves were included. This can be explained by the fact that the prevalence of MRSA in pigs was highest (89.3%), low in dairy cattle (6.2%) and <0.05% in broilers (NethMap/MARAN 2022, http://hdl.handle.net/10029/625885). Since 2000 the *cfr* gene has been of reported with increasing frequency to be present in coagulase-negative staphylococcal and *Mammaliicoccus* species, obtained from animals and humans and in *S. aureus* from animals[2,17–22]. However, reports on *cfr*-positive MRSA isolated from humans are scarce. The first reports on *cfr* in MRSA originate from the USA and date from 2007 and 2008[23,24]. In 2010 there was a report on an outbreak with *cfr*-positive MRSA that occurred in an intensive care unit in Spain involving 12 patients, showing nosocomial transmission of such strains do occur[25]. Since then, a limited number of papers have been published on *cfr* in MRSA with various genetic backgrounds[26–38]. In our study all *cfr*-positive isolates were CC398 and to our knowledge only two other groups, from Spain and Belgium, reported on *cfr*-positive CC398 obtained from humans[39,40]. In the Netherlands ~25% of the MRSA cultured from humans are CC398. Furthermore, the CC398 carriage in livestock and in particular pigs is still very high in the Netherlands, despite the reduction of the use of antibiotics in livestock by nearly 70%, as compared to the reference year 2009 (Report of the Netherlands Veterinary Medicines Institute (SDa): Usage of antibiotics in agricultural livestock in the Netherlands in 2020 (https://www.autoriteitdiergeneesmiddelen.nl/en/)[41]. In livestock, tetracycline is the most frequently used antibiotic and as almost all CC398 are tetracycline resistant, this usage selects for CC398 [NethMap/MARAN 2022, http://hdl.handle.net/10029/625885]. Oxazolidinones and streptogramin A are not used in livestock, but lincosamides, pleuromutilins and florfenicol are, which may select for *cfr*. The restricted use of oxazolidinones in humans and the frequent use of pleuromutilins, florfenicol and lincosamides in livestock and the finding that *cfr* was present in CC398, but not in non-CC398, suggests that *cfr* in MRSA in the Netherlands originates from the animal reservoir. Two earlier reports on *cfr* in CC398 from Spain and Belgium also were from a pig farmer and from a person with professional contact with pigs and cows[39,40]. A recent study on linezolid-resistant isolates from food-producing animals in Belgium included the analysis of six ST398 *S. aureus* isolates that all carried the *cfr* gene[42]. These studies support the hypothesis that the *cfr* gene likely originates from animals. The composition of the plasmids identified in this study was diverse and the plasmids were found in genetically distinct CC398 isolates. This shows that there is no dissemination of one or more *cfr*-positive strains or of particular *cfr*-carrying plasmids. In seven of the eight isolates the *cfr*-gene was located in the Tn*558* transposon and therefore linked to the *fexA* gene, suggesting that spread of the *cfr* gene may have occurred via transposition of the Tn*558*. These findings suggest that there have been multiple introductions of *cfr* in the MRSA population in the Netherlands. The source of the *cfr* gene remains unclear, but as it has been found in many different *Staphylococcus* species and species from other genera e.g., *Enterococcus* there is a very large potential reservoir[14].

The *fexA* gene in our collection was found to be highly variable with 30 sequence variants resulting in a Simpson's diversity index of ~0.81. Interrogation of the NBI database revealed that the high variation was not restricted to *Staphylococcus* species. It is unclear

what the reason for this extensive sequence variation is. Possibly the *fexA* gene is under selective pressure due to extensive antibiotic use or adaptation when the gene is transferred from one species to another. There are only very few reports on *fexA* sequence variation. In 2013 Gomez-Sanz et al. reported on a *fexA* variant in a chloramphenicol resistant canine *S. pseudintermedius* that did not confer florfenicol resistance and in 2016 a florfenicol susceptible MRSA isolated from meat in Germany had a similar altered *fexA* gene[43,44]. Recently, Müller et al. reported that some mutations in the synthetically created *fexA* gene lowered resistance levels for chloramphenicol and florfenicol which they substantiated by protein modeling[45]. We determined MICs for chloramphenicol and florfenicol for six of the *fexA* variants found in our collection. This showed that isolates carrying four variants were resistant to both chloramphenicol and florfenicol, but those carrying two *fexA* variants were resistant to chloramphenicol but sensitive for florfenicol. These two variants *fexA17* and *fexA28* were found exclusively in CC5 MRSA and not in CC398. In the Netherlands, CC5 MRSA is only rarely found in animals. Florfenicol is not used for treatment of human patients but is used for treatment of livestock and may have selected for the observed *fexA* encoded florfenicol resistance. The effects of the *fexA* sequence variation will be subject for further study.

In conclusion, we have shown that the multidrug-resistance gene *cfr* and the chloramphenicol- and florfenicol-resistance gene *fexA* are present in MRSA isolates from humans and animals in the Netherlands. The *cfr* gene was found exclusively in CC398 and *fexA* predominantly in CC398 and in the CC5. The proportion of *cfr*-positive MRSA is low, and the reserve antibiotic linezolid is rarely used in the Netherlands, yet its presence is worrying and should be closely monitored.

## Data availability

All chromosome and plasmid sequences have been deposited in the NCBI database. Accession numbers can be found in the Methods section. Patient's data, data on health-care centers and data on farms are not available because of privacy and confidentiality reasons. All assembled plasmids and chromosomes are deposited in the NCBI database under the following accession numbers: H1_RIVM_M044329: CP096540-CP096541, H2_RIVM_M047065: CP096539, H3_RIVM_M047916: CP096535-CP096538, H4_RIVM_M083782: CP096532-CP096534, H5_RIVM_M084526: CP096530-CP096531, H6_RIVM_M084986: CP096528-CP096529, H7_RIVM_M087195: CP096526-CP096527, P1_RIVM_M085090: CP096522-CP096525. Additional data or materials are available from Engeline van Duijkeren (Engeline.van.Duijkeren@rivm.nl).

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

## Acknowledgements

This research was funded by the Dutch Ministry of Health, Welfare and Sport (V/150302/22/BR and V/150166/21/SV) and the Ministry of Agriculture Nature and Food Quality grant BO-43-111-010.

## Author contributions

L.M.S., K.V. and E.v.D. conceived and designed the study. P.H., B.Wu., M.R. and B.Wi. collected specimens and cultured the MRSA isolates from the livestock sampling. S.W. and M.v.S.-V. assembled next-generation sequence data. F.L. performed long-read sequencing and reconstruction of the plasmids and genomes. K.V. and M.S.M.B. performed phenotypic resistance testing. C.D. and A.P.A.H. critically reviewed the manuscript. All members of Dutch MRSA surveillance study group received and reviewed the manuscript. L.M.S. performed analyses of the molecular data and wrote the paper.

## Competing interests

The authors declare no competing interests.

## Additional information

## the Dutch MRSA surveillance study group

A. Maijer-Reuwer[6], M. A. Leversteijn-van Hall[7], W. van den Bijllaardt[8], R. van Mansfeld[9], K. van Dijk[10], B. Zwart[11], B. M. W. Diederen[12], J. W. Dorigo-Zetsma[13], D. W. Notermans[14], A. Ott Certe[15], W. Ang[16], J. da Silva[17], A. L. M. Vlek[18], A. G. M. Buiting[19], L. Bode[20], S. Paltansing[21], A. J. van Griethuysen[22], M. den Reijer[23], M. J. C. A. van Trijp[24], M. Wong[25], A. E. Muller[26], M. P. M. van der Linden[27], M. van Rijn[28], S. B. Debast[29], K. Waar[30], E. Kolwijck[31], N. Alnaiemi[32], T. Schulin[33], S. Dinant[34], S. P. van Mens[35], D. C. Melles[36], J. W. T. Cohen Stuart[37], P. Gruteke[38], I. T. M. A. Overdevest[39], A. van Dam[40], I. Maat[41], B. Maraha[42], J. C. Sinnige[43], E. E. Mattsson[44], M. van Meer[45], A. Stam[46], E. de Jong[47], S. J. Vainio[48], E. Heikens[49], R. Steingrover[50], A. Troelstra[51], E. Bathoorn[52], T. A. M. Trienekens[53], D. W. van Dam[54], E. I. G. B. de Brauwer[55] & H. Berkhout[56]

[6]ADRZ medisch centrum, Department of Medical Microbiology, Goes, Netherlands. [7]Alrijne Hospital, Department of Medical Microbiology, Leiden, Netherlands. [8]Amphia Hospital, Microvida Laboratory for Microbiology, Breda, Netherlands. [9]Amsterdam UMC—location AMC, Department of Medical Microbiology, Amsterdam, Netherlands. [10]Amsterdam UMC—location Vumc, Department of Medical Microbiology and Infection Control, Amsterdam, Netherlands. [11]Atalmedial, Department of Medical Microbiology, Amsterdam, Netherlands. [12]Bravis Hospital/ZorgSaam Hospital Zeeuws-Vlaanderen, Department of Medical Microbiology, Roosendaal/Terneuzen, Netherlands. [13]CBSL, Department of Medical Microbiology, Hilversum, Netherlands. [14]Centre for Infectious Disease Control, National Institute for Public Health and the Environment, Bilthoven, Netherlands. [15]Department of Medical Microbiology, Groningen, Netherlands. [16]Department of Medical Microbiology, Hoorn, Netherlands. [17]Deventer Hospital, Department of Medical Microbiology, Deventer, Netherlands. [18]Diakonessenhuis, Department of Medical Microbiology and Immunology, Utrecht, Netherlands. [19]Elisabeth-TweeSteden (ETZ) Hospital, Department of Medical Microbiology and Immunology, Tilburg, Netherlands. [20]Erasmus University Medical Center, Department of Medical Microbiology, Rotterdam, Netherlands. [21]Franciscus Gasthuis & Vlietland, Department of Medical Microbiology and Infection Control, Rotterdam, Netherlands. [22]Gelderse Vallei Hospital, Department of Medical Microbiology, Ede, Netherlands. [23]Gelre Hospitals, Department of Medical Microbiology and Infection prevention, Apeldoorn, Netherlands. [24]Groene Hart Hospital, Department of Medical Microbiology and Infection Prevention, Gouda, Netherlands. [25]Haga Hospital, Department of Medical Microbiology, 's-Gravenhage, Netherlands. [26]HMC Westeinde Hospital, Department of Medical Microbiology, 's-Gravenhage, Netherlands. [27]IJsselland hospital, Department of Medical Microbiology, Capelle a/d IJssel, Netherlands. [28]Ikazia Hospital, Department of Medical Microbiology, Rotterdam, Netherlands. [29]Isala Hospital, Laboratory of Medical Microbiology and Infectious Diseases, Zwolle, Netherlands. [30]Certe Medische Microbiologie Friesland | Noordoostpolder, Department of Medical Microbiology, Leeuwarden, Netherlands. [31]E. Kolwijck, Jeroen Bosch Hospital, Department of Medical Microbiology and Infection Control, 's-Hertogenbosch, Netherlands. [32]LabMicTA, Regional Laboratory of Microbiology Twente Achterhoek, Hengelo, Netherlands. [33]Laurentius Hospital, Department of Medical Microbiology, Roermond, Netherlands. [34]Maasstad Hospital, Department of Medical Microbiology, Rotterdam, Netherlands. [35]Maastricht University Medical Centre, Department of Medical Microbiology, Maastricht, Netherlands. [36]Meander Medical Center, Department of Medical Microbiology, Amersfoort, Netherlands. [37]Noordwest Ziekenhuisgroep, Department of Medical Microbiology, Alkmaar, Netherlands. [38]OLVG Lab BV, Department of Medical Microbiology, Amsterdam, Netherlands. [39]PAMM, Department of Medical Microbiology, Veldhoven, Netherlands. [40]Public Health Service, Public Health Laboratory, Amsterdam, Netherlands. [41]Radboud University Medical Center, Department of Medical Microbiology, Nijmegen, Netherlands. [42]Albert Schweitzer Hospital, Department of Medical Microbiology, Dordrecht, Netherlands. [43]Regional Laboratory of Public Health, Department of Medical Microbiology, Haarlem, Netherlands. [44]Reinier de Graaf Groep, Department of Medical Microbiology, Delft, Netherlands. [45]Rijnstate Hospital, Laboratory for Medical Microbiology and Immunology, Velp, Netherlands. [46]Saltro Diagnostic Centre, Department of Medical Microbiology, Utrecht, Netherlands. [47]Slingeland Hospital, Department of Medical Microbiology, Doetinchem, Netherlands. [48]St. Antonius Hospital, Department of Medical Microbiology and Immunology, Nieuwegein, Netherlands. [49]St. Jansdal Hospital, Department of Medical Microbiology, Harderwijk, Netherlands. [50]St. Maarten Laboratory Services, Department of Medical Microbiology, Cay Hill, St. Maarten.

[51]University Medical Center Utrecht, Department of Medical Microbiology, Utrecht, Netherlands. [52]University of Groningen, Department of Medical Microbiology, Groningen, Netherlands. [53]VieCuri Medical Center, Department of Medical Microbiology, Venlo, Netherlands. [54]Zuyderland Medical Centre, Department of Medical Microbiology and Infection Control, Sittard-Geleen, Netherlands. [55]Zuyderland Medical Centre, Department of Medical Microbiology and Infection Control, Heerlen, Netherlands. [56]Canisius Wilhelmina Hospital, Department of Medical Microbiology and Infectious Diseases, Nijmegen, Netherlands.

