## [Peer Review File · Communications Medicine]

Reviewers' comments:

Reviewer #1 (Remarks to the Author):

This study by Schouls et al. assessed potential transmission of MRSA between human and animal hosts using >6500 MRSA isolates. The authors identified eight livestock-associated MRSA (LA-MRSA) strains carrying *cf*r gene that confers the PhLOPS multi-resistance phenotype. In addition to the *cf*r gene, prevalence of *fexA* gene and the heterogeneity of *fexA* ORF were identified in the MRSA isolates. The presented work is solid but I believe the manuscript could be strengthened by more data on the functional role of the observed sequence variations in *fexA* genes.

My only concern with the manuscript is:

-The authors identified ~30 different sequence variations in the *fexA* gene resulting 27 allelic variations in the FexA protein. Some of the major *fexA* variants, such as *fexA03*, *fexA05*, and *fexA17* can be further analyzed for their role in chloramphenicol and florfenicol resistance phenotypes.

Reviewer #2 (Remarks to the Author):

Thank you very much for the opportunity to review this important article, in which the authors characterized MRSA from humans and livestock in the Netherlands with an emphasis on *cf*r and *fexA* positive strains.

I have the following comments and suggestions:

Summary:

The two sentences

“6,327 MRSA isolates collected from humans in the Dutch national surveillance and 3 332 livestock-associated MRSA (LA-MRSA)” vs. “Of the surveillance isolates 3.9% carried *fexA* and 7.5% of the livestock samples carried *fexA*”

... seem to be contractive (if only reading the abstract), as only 8 isolates with *cf*r were included, but 7.5% of 6327 would be 475 isolates. Please clarify.

Introduction:

L17

“For this reason, national surveillance of MRSA in humans...” do the authors mean national surveillance or national molecular surveillance? Wasn't a national surveillance introduced before 1989 (as preventive measures and S/D strategy in NL started earlier?) Please clarify.

L20f.

Henceforth, the term LA-MRSA is used for all MRSA CC398 isolates, irrespectively of whether the isolate was hospital-acquired or indeed connected to livestock contact. This should be stated clearly; either here or in the method section. An alternative could be to consider using the term MRSA CC398 instead of LA-MRSA?

L24

“...with prevalence ranging from 22% to 43%.” Please correct. The proportion of resistant isolates among all MRSA isolates is not a “prevalence” (which refers to a proportion within a population tested)

L26ff.

The authors should consider adding a short sentence that cfr positive isolates are also linezolid-resistant (as relevant for surveillance and clinical impact) and shortly explain the PhLOPSA phenotype here in order to improve the description of the study question.

Methods:

In several publications, the occurrence of the *optrA* gene was also described, mainly in Chinese isolates (mostly enterococci, but also *S. aureus* and non-aureus staphylococci). This gene is very similar to cfr regarding the resulting resistance phenotype. As whole genome data are available, the authors should consider analyzing the presence of *optrA* in the isolates as well, because this would greatly increase the value of the manuscript. If not possible, this could / should be discussed (see comment on limitations below).

L92ff. I think it might be easier just to state concretely which cut-off MICs were used for the PhLOPSA antibiotics.

Results:

The reader wonders whether it could be shown how many persons from whom MRSA isolates were characterized, had contact with pigs, cattle, poultry or other livestock. The results suggest that this was assessed? This would provide some information on the question whether or not MRSA CC398 is still epidemiologically “livestock-associated” or more like a Dutch CA-MRSA strain.

Is there further information on the humans from whom the isolates were obtained? Did they receive linezolid-therapies or clindamycin?

Can the authors add the *spa* types for the cfr positive isolates (in fig. 2 or the text) and the *spa* types and MLST STs (instead of genogroups) for the *fexA* strains?

Discussion:

L288f. To help the reader understand the numbers cited, the authors could also mention the total DDD/population or total DDD/patient-days that are prescribed (this would demonstrate that clindamycin accounts only for a small proportion of all DDDs).

L297f. I am struggling with the sentence “the emergence of cfr-positive MRSA MRSA is worrying, but not yet alarming” for two reasons: Isn’t another major indication for linezolid use in the Netherlands the oral (prolonged) treatment of severe biofilm-associated infections due to coagulase-negative staphylococci (e.g. prosthetic joint infections), as these are often methicillin-resistant? While for i.v. Treatment there are some alternatives, linezolid is one of the rare “oral” options to treat VRE and MR-staphylococcal infections (especially if levofloxacin and doxycycline are resistant, which is not rare in MRSA). This could be mentioned.

L302ff. Can the authors add some data how often drugs that specifically select for cfr (e.g. fenicolis etc.) are used in Dutch livestock? This could stress the argument given in line 322f. The problem is that amoxicillin, tetracyclines etc. also select for MRSA, which are among the most frequently prescribed antimicrobials both in humans and veterinary medicine. However, I agree that the specific selection pressure for this gene due to PhLOPSA antibiotics is very low among humans (as hardly any VRE and MRSA infections occur).

It should be discussed that the 333 isolates from animals in the supplementary table included only 14 from farmers. Hence, occupational exposure is not analyzed in this manuscript. Is there any data regarding this issue from other currently performed studies? I think this would be worth to discuss as former investigations have shown very frequent colonization of directly exposed farmers. If the isolates were from epidemiologically unrelated farms, this indicates that these strains are quite widespread and in consequence the potential for spread among farmers is likely.

A limitation of the study, might also be that the isolates from livestock comprised mostly porcine isolates and very few from cattle, which, I think, have a higher florfenicol exposure in some countries (see a paper by Cuny C; *Vet Microbiol.* 2017 Feb;200:88-94 who describe occurrence of cfr in nasal isolates from farmers)?

L322f. Is there data about the emergence of cfr in non-aureus staphylococci or *M. sciuri* on Dutch livestock farms? This would support the discussion about the origin, as These strains could be a reservoir for repeated introduction of cfr into MRSA CC398. It could be discussed whether it would make sense to plan such studies, if not yet done.

Response to the reviewers' comments

Reviewer #1 (Remarks to the Author):

This study by Schouls et al. assessed potential transmission of MRSA between human and animal hosts using >6500 MRSA isolates. The authors identified eight livestock-associated MRSA (LA-MRSA) strains carrying cfr gene that confers the PhLOPS multi-resistance phenotype. In addition to the cfr gene, prevalence of fexA gene and the heterogeneity of fexA ORF were identified in the MRSA isolates. The presented work is solid but I believe the manuscript could be strengthened by more data on the functional role of the observed sequence variations in fexA genes.

Response: We thank the reviewer for the positive response to our work.

My only concern with the manuscript is:

The authors identified ~30 different sequence variations in the fexA gene resulting 27 allelic variations in the FexA protein. Some of the major fexA variants, such as fexA03, fexA05, and fexA17 can be further analyzed for their role in chloramphenicol and florfenicol resistance phenotypes.

Response: We agree with the reviewer that the sequence variation of the fexA gene is interesting, and we already mentioned that we will study this in detail in upcoming studies. However, in the meantime we have performed a pilot by determining the MICs of 24 isolates, comprising six fexA alleles. Eighteen isolates were obtained from the human surveillance collection and six from the livestock sampling. The analyses showed that all isolates were resistant to both chloramphenicol and florfenicol except for isolates carrying fexA17 and fexA28. Latter isolates were resistant to chloramphenicol but sensitive for florfenicol. The fexA17 and fexA28 alleles were only found among the CC5 isolates. We have now added the results of this pilot at the end of the Results and Discussion sections of the manuscript.

Reviewer #2 (Remarks to the Author):

Thank you very much for the opportunity to review this important article, in which the authors characterized MRSA from humans and livestock in the Netherlands with an emphasis on cfr and fexA positive strains. I have the following comments and suggestions:

Response: We wish to thank the reviewer for the positive and constructive review of our manuscript.

Summary:

The two sentences

"6,327 MRSA isolates collected from humans in the Dutch national surveillance and 3 332 livestock-associated MRSA (LA-MRSA)" vs. "Of the surveillance isolates 3.9% carried fexA and 7.5% of the livestock samples carried fexA"

... seem to be contractive (if only reading the abstract), as only 8 isolates with cfr were included, but 7.5% of 6327 would be 475 isolates. Please clarify.

Response: We agree with the reviewer that this may be confusing. Therefore, we have changed the text in the Abstract from 'All cfr-positive isolates also carried the phenicol-resistance gene fexA ...' into 'Of the human surveillance isolates 3.9% carried the phenicol-resistance gene fexA as did 7.5% of the isolates obtained from livestock-related samples. All cfr-positive isolates also carried fexA.'

Introduction:

L17.

“For this reason, national surveillance of MRSA in humans...” do the authors mean national surveillance or national molecular surveillance? Wasn't a national surveillance introduced before 1989 (as preventive measures and S/D strategy in NL started earlier?) Please clarify.

Response: The Dutch national MRSA surveillance was introduced in 1989. The Dutch Health Inspectorate urged all MML's to submit their isolates to the RIVM for further analysis, making it essentially mandatory. In 1989 phage-typing was used to characterize the isolates, which was replaced by PFGE and later by spa-typing. From 2008 onward MLVA was performed next to spa-typing and in 2015 spa-typing was abandoned and isolates were typed by MLVA only. Currently, surveillance comprises molecular characterization of the isolates by MLVA and NGS. In addition, epidemiological data for risk assessment is collected via the Type-Ned MRSA system. In a separate system called ISIS-AR, phenotypic antibiotic resistance data of all bacterial species that are generated by the MMLs is collected and aggregated for reporting. Both Type-Ned and ISIS-AR are part of the Dutch MRSA surveillance.

L20f.

Henceforth, the term LA-MRSA is used for all MRSA CC398 isolates, irrespectively of whether the isolate was hospital-acquired or indeed connected to livestock contact. This should be stated clearly; either here or in the method section. An alternative could be to consider using the term MRSA CC398 instead of LA-MRSA?

Response: We agree with the reviewer that the term LA-MRSA is confusing. Per definition LA-MRSA would be all MRSA acquired from or associated with livestock. To avoid confusion, we have replaced LA-MRSA by CC398 throughout the text and indicated whether these CC398 isolates were obtained from humans or animals. Also, we have replaced the genogroup designations by clonal complexes as data on the wgMLST genogroup assignment have not yet been published.

L24.

“...with prevalence ranging from 22% to 43%.” Please correct. The proportion of resistant isolates among all MRSA isolates is not a “prevalence” (which refers to a proportion within a population tested)

Response: We agree and have replaced 'prevalence' with 'proportion' where appropriate throughout the text.

L26ff.

The authors should consider adding a short sentence that cfr positive isolates are also linezolid-resistant (as relevant for surveillance and clinical impact) and shortly explain the PhLOPSA phenotype here in order to improve the description of the study question.

Response: Although we think the description of the PhLOPS_A phenotype fits very well in the Results section, we do understand the reviewer's reasoning and therefore we have now also included this in the Introduction.

Methods:

*In several publications, the occurrence of the *optrA* gene was also described, mainly in Chinese isolates (mostly enterococci, but also *S. aureus* and non-aureus staphylococci). This gene is very similar to *cfr* regarding the resulting resistance phenotype. As whole genome data are available, the authors should consider analyzing the presence of *optrA* in the isolates as well, because this would greatly increase the value of the manuscript. If not possible, this could / should be discussed (see comment on limitations below).*

Response: We agree with the reviewer that the *optrA* gene is also an important rare antibiotic gene. Indeed, in the current selection of isolates we did find only one CC398 isolate in the Dutch MRSA surveillance collection carrying this gene. This *optrA* variant yields a borderline MIC for linezolid. We are currently preparing a separate short report on *optrA* en *poxxA* in isolates obtained in the Dutch MRSA surveillance. For this reason, we do not incorporate the *optrA* finding in the current manuscript.

L92ff. I think it might be easier just to state concretely which cut-off MICs were used for the PhLOPSA antibiotics.

Response: The cut-off MICs have been incorporated in Table 1 as values in parenthesis. We believe having these values available in the table rather than in the Materials & Methods section is much more informative and better helps to interpret the results.

Results:

The reader wonders whether it could be shown how many persons from whom MRSA isolates were characterized, had contact with pigs, cattle, poultry or other livestock. The results suggest that this was assessed? This would provide some information on the question whether or not MRSA CC398 is still epidemiologically “livestock-associated” or more like a Dutch CA-MRSA strain.

Response: It is well known that contact with livestock is the major risk factor for humans to get colonized with CC398 MRSA. In the Dutch MRSA surveillance, risk factors are assessed via questionnaires that are filled out by MMLs or their affiliated infection prevention employees. For the sequenced isolates (2019-2021), questionnaires were completed for 85% of all patients and 13% of these patients reported to have been in contact with livestock. Of persons carrying CC398 50% reported having contact with livestock. In 90% of all persons reporting livestock contact, the MRSA were CC398. We have now added this information in the Materials and Methods section on Metadata.

The wgMLST analysis (Figure 2) showed there is not a single Dutch LA-MRSA strain. In fact, there was a considerable degree of diversity among the CC398 isolates with an average distance of 171 alleles. In comparison the average allelic distance within CC5 isolates is 321 and in CC6 127. As isolates differing no more than 16 alleles in wgMLST are suspect of representing the same strain this shows many LA-MRSA strains are circulating in the Netherlands.

Is there further information on the humans from whom the isolates were obtained? Did they receive linezolid-therapies or clindamycin?

Response: Unfortunately, we do not have clinical data or data on antibiotic treatment of the persons from whom we obtained MRSA isolates.

Can the authors add the spa types for the cfr positive isolates (in fig. 2 or the text) and the spa types and MLST STs (instead of genogroups) for the fexA strains?

Response: We have not been performing spa-typing since 2015. Therefore, we cannot provide information on spa-types. Also, spa-typing, classical MLST and MLVA do not provide very useful information, because of their limited discriminatory power for CC398 isolates. The NGS data of the cfr-positive and fexA-positive isolates is much more informative.

However, to provide the reviewer and the readers with information on the MLST we have replaced S-Table 5. in the Supplementary data file with a new version that includes the STs.

Discussion:

L288f. To help the reader understand the numbers cited, the authors could also mention the total DDD/population or total DDD/patient-days that are prescribed (this would demonstrate that clindamycin accounts only for a small proportion of all DDDs).

Response: In NethMap an inpatient use of systemic antibiotics 85.79 DDD/100 patient-days in 2019 is reported. We have added the inpatient and outpatient usage to the text.

L297f. I am struggling with the sentence “the emergence of cfr-positive MRSA MRSA is worrying, but not yet alarming” for two reasons: Isn’t another major indication for linezolid use in the Netherlands the oral (prolonged) treatment of severe biofilm-associated infections due to coagulase-negative staphylococci (e.g. prosthetic joint infections), as these are often methicillin-resistant? While for i.v. Treatment there are some alternatives, linezolid is one of the rare “oral” options to treat VRE and MR-staphylococcal infections (especially if levofloxacin and doxycycline are resistant, which is not rare in MRSA). This could be mentioned.

Response: We do agree that combination of the terminology ‘worrying’ and ‘not yet alarming’ is confusing. For this reason, we have changed the text in the Discussion, removing ‘not alarming’.

Linezolid consumption in the Netherlands is extremely low at 0.08 DDD/100 patient-days in hospitals in 2019 (NethMap) since it is a last resort antibiotic. It may be used for treatment of infections vancomycin-resistant micro-organisms if no other antibiotic is available. Also, it may be used for treatment community-acquired pneumonia and complicated infections of the skin and soft tissue if other treatment regimens fail. Lastly, linezolid may be used to treat patients infected with MDR-TB or XDR-TB, infections that are rare in the Netherlands. A recent paper by Zijlstra *et al.* (Arthroplasty volume 4, Article number: 19 (2022)) on a protocol for treatment of periprosthetic joint infections, antibiotic treatment regimens did not include linezolid. In our Discussion we have made several remarks on the use of linezolid in the Netherlands e.g., in VRE and MDR-MRSA and we believe we should not expand this any further. Due to the low usage of linezolid in the Netherlands there is no reason for drastic measures, but the emergence of cfr-positive MRSA is worrying.

L302ff. Can the authors add some data how often drugs that specifically select for cfr (e.g. fenicols etc.) are used in Dutch livestock? This could stress the argument given in line 322f. The problem is that amoxicillin, tetracyclines etc. also select for MRSA, which are among the most frequently prescribed antimicrobials both in humans and veterinary medicine. However, I agree that the specific selection pressure for this gene due to PhLOPSA antibiotics is very low among humans (as hardly any VRE and MRSA infections occur).

Response: We have inserted the reference to the Netherlands Veterinary Medicines Institute (SDa) monitoring on antimicrobial consumption in the animals in the Netherlands. Here numbers on antibiotic usage in animal husbandry can be found. We added two sentences on the use of phenicols in Dutch livestock. These sentences read:

In the Netherlands, phenicols are mainly used in pigs and veal calves, and only small amounts are used in dairy cattle and no use is reported in poultry (Report of the Netherlands Veterinary Medicines Institute (SDa): Usage of antibiotics in agricultural livestock in the Netherlands in 2020 (<https://www.autoriteitdiergeenemiddelen.nl/en/>)). In the present study, most isolates from animals were from pigs, and much less from dairy cattle and poultry, no isolates from veal calves were included. This can be explained by the fact that the prevalence of MRSA in pigs was highest (89.3%), low in dairy cattle (6.2%) and less than 0.05% in broilers (NethMap/MARAN 2022, DOI: [10.21945/RIVM-2022-0057](https://doi.org/10.21945/RIVM-2022-0057)).

It should be discussed that the 333 isolates from animals in the supplementary table included only 14 from farmers. Hence, occupational exposure is not analyzed in this manuscript. Is there any data regarding this issue from other currently performed studies? I think this would be worth to discuss as former investigations have shown very frequent colonization of directly exposed farmers. If the isolates were from epidemiologically unrelated farms, this indicates that these strains are quite widespread and in consequence the potential for spread among farmers is likely.

Response: We agree this is an important question. However, our study is on the occurrence and characteristics of MRSA carrying the *cfr* and *fexA* genes found in the human population in the Netherlands and in livestock animals and on animal farms. It is not on the transmission of MRSA found in animals on farms to humans working on these farms. A completely different study design would have been required. Our study shows that we found seven *cfr*-positive CC398 MRSA in collection of the Dutch MRSA surveillance isolates. Four of these humans reported to have been in contact with livestock, one reported not to have been in contact and for two persons this was unknown.

There are many studies on transmission of LA-MRSA to humans, but unfortunately, most of these do not specifically investigate *cfr* or *fexA* genes. We also have previously performed and reported two studies on occupational exposure: Bosch, T., *et al.* Transmission and persistence of livestock-associated methicillin-resistant *Staphylococcus aureus* among veterinarians and their household members. *Appl Environ Microbiol* 81, 124-129 (2015); Verkade, E., *et al.* Transmission of methicillin-resistant *Staphylococcus aureus* CC398 from livestock veterinarians to their household members. *PLoS One* 9, e100823 (2014). In our current study, four of the 14 persons carried a MRSA that was indistinguishable from isolates obtained from animals on the farm

where they were working. This may suggest transmission from animal to human, but epidemiological data to substantiate this are lacking.

We disagree with the reviewer that we need to discuss the data in Table any further. However, the reviewer addresses a valid point on the description of the humans in the livestock sampling. We therefore changed the description in the Materials & Methods section from ‘... nasal swabs from farmers ...’ into ‘... nasal swabs from persons working on these farms ...’. Also, the table in the supplementary data contained 333 isolates, we have corrected this and replaced the table with the correct number of isolates (n=332).

A limitation of the study, might also be that the isolates from livestock comprised mostly porcine isolates and very few from cattle, which, I think, have a higher florfenicol exposure in some countries (see a paper by Cuny C; Vet Microbiol. 2017 Feb;200:88-94 who describe occurrence of cfr in nasal isolates from farmers)?

Response: We do agree with the reviewer that the livestock sampling was dominated by porcine samples (67%) and only few isolates from dairy cattle were included (7%). The reason is that the MRSA-carriage in the Netherlands is very high in pigs and low in dairy cattle. We hope that the systematic annual sampling of animals that started in 2018 will yield more isolates from cattle (including veal calves) and poultry to provide a more balanced collection.

It is likely that use of florfenicol will select *cfr* and *fexA* carrying MRSA. The use of amphenicols is higher in pigs, compared to that in dairy cattle and it is not used in poultry. The use is highest in veal calves, but no isolates from veal calves were included to date. We monitor one animal sector each year and currently (2022) we are monitoring veal calves, so in the future we will be able to see if MRSA carrying *cfr* are present in this animals sector. We added this information to the Discussion. It now reads:

In the present study, most isolates from animals were from pigs, and much less from dairy cattle and poultry, no isolates from veal calves were included. This can be explained by the fact that the prevalence of MRSA in pigs was highest (89.3%), low in dairy cattle (6.2%) and very low (0.05%) in broilers (NethMap/MARAN 2022, DOI: [10.21945/RIVM-2022-0057](https://doi.org/10.21945/RIVM-2022-0057)).

L322f. Is there data about the emergence of cfr in non-aureus staphylococci or M. sciuri on Dutch livestock farms? This would support the discussion about the origin, as These strains could be a reservoir for repeated introduction of cfr into MRSA CC398. It could be discussed whether it would make sense to plan such studies, if not yet done.

Response: We don't have data on *cfr* in CoNS in Dutch farms and therefore cannot add this to the discussion. As no surveillance for CoNS or MSSA exists, we don't know if and how frequent *cfr*-positive MSSA and CoNS circulate in the Dutch human and livestock population. However, NethMap that annually reports on antibiotic resistance (ISIS-AR), shows that linezolid resistance for outpatients and inpatients is virtually non-existent. Also, no linezolid resistance is reported for *E. faecalis* and *E. faecium*. We agree that it is feasible that CoNS in the Netherlands may act as a reservoir for the *cfr* gene, but without any supporting data discussion on this would be pure speculation.

REVIEWERS' COMMENTS:

Reviewer #1 (Remarks to the Author):

The revised manuscript has been improved according to the suggestions of the reviewers. I would recommend 'Accept' for the revised manuscript.

Reviewer #2 (Remarks to the Author):

The comments have been addressed.